# Author Identification Using Chaos Game Representation and Deep Learning

**Catalin Stoean** [1,2,*] and **Daniel Lichtblau** [3]

1    Human Language Technology Research Center, University of Bucharest, 010014 Bucharest, Romania
2    Grupo Ingeniería de Sistemas Integrados, E.T.S.I. Telecomunicación, Universidad de Málaga, 29071 Málaga, Spain
3    Wolfram Research, Champaign, IL 61820, USA; danl@wolfram.com
\*    Correspondence: catalin.stoean@fmi.unibuc.ro

**Abstract:** An author unconsciously encodes in the written text a certain style that is often difficult to recognize. Still, there are many computational means developed for this purpose that take into account various features, from lexical and character-based attributes to syntactic or semantic ones. We propose an approach that starts from the character level and uses chaos game representation to illustrate documents like images which are subsequently classified by a deep learning algorithm. The experiments are made on three data sets and the outputs are comparable to the results from the literature. The study also verifies the suitability of the method for small data sets and whether image augmentation can improve the classification efficiency.

**Keywords:** authorship attribution; chaos game representation; deep learning

## 1. Introduction

The style of every author is encoded into the documents that the person has written, be that these are books, articles, or simply emails or some small statement in a social network. The style gets more consistent as the amount of text written is larger. The authorship attribution (AA) task assumes that there is a training set of documents for which the authors are known and another test set of texts where the writer is not acknowledged in advance but these are, in general, written by one of the authors from the training samples. The goal is to determine who is the author for each of the test samples. Most computational approaches deal with counting of various components in the texts for determining the style of the writer and, accordingly, the author. A good survey for such methods can be read in Reference [1] or in an overview for a competition in AA [2].

The method put forward in the current research suggests a solution that is very different from the standard approaches. While in general the trend is to transform images to text to get their meaning, like in Reference [3], herein we aim to encode the entire text in one image. To provide a simple and intuitive overview, the present methodology proposes that the text documents are substituted by a chaos game representation (CGR) and the obtained images subsequently fed to a deep learning (DL) method to learn specific characteristics for each author from the obtained illustrations. The DL could next distinguish between similar representations that result from new test documents which are also transformed via CGR. In a previous proposed approach [4], CGR proved to be efficient for encoding the style of the writers with the same goal of determining the authors, but using a shallow classifier, and not the indisputable power of DL for image processing. However, the latter comes with a high demand, that of supplying a large amount of data to train on, which is not always available. Nevertheless, the goal of the current article is to study the suitability of the CGR-DL tandem for the AA task and there are two benchmark problems that are considered for experimentation.

We formulate the following research questions for the current work:

1. Is DL appropriate for extracting meaningful information from the CGR representations of the text documents and using it to distinguish between the candidate authors?
2. Does DL require a high amount of CGR images to reach results that are competitive with respect to the state of the art?
3. Is it possible to artificially generate augmented images and use them to improve accuracy for the data sets where the number of samples is small?

Next section will present additional information about the AA problem and will also briefly describe some approaches that perform well for this task. Section 3 will introduce the proposed CGR-DL tandem. Experimental results are described in section 4 and therein the research questions will also be answered. Section 5 presents some final remarks and ideas of taking the research further.

## 2. Materials

Many of the benchmark data sets have a preset separation into training and test samples and it often happens that the two sets are equal. In order to find a proper DL model however, a validation set is also required. The DL model that contains the appropriate values for the training-validation data is often chosen to be the one that leads to the highest validation accuracy. This is subsequently used for classifying the test samples. One such data set is the CCAT10, originally proposed in Reference [5], that contains 10 authors, each with 50 articles in the training set and the same authors with a distinct set of 50 articles in the test set. This data set is considered with the aim of answering the second research question from the introduction.

A larger data set that is considered in the current study is one containing reviews for movies from users registered at http://www.imdb.com. The data set is commonly referred to as IMDb62, it contains 62,000 reviews that are written by 62 users, each author of 1000 reviews. The data set was put forward in Reference [6].

The third considered test case is represented by a topic that got the attention of the public because it appeared in the international press. The name of the author of *The Cuckoo's Calling* novel was discovered by means of computers using computational methods for AA. The novel was published under the pseudonym Robert Galbraith and it was revealed later that the actual author was J.K. Rowling (famous especially for the Harry Potter series). We propose a similar scenario where seven candidates are considered, each with a novel from the crime and mystery genre. The ones considered are further enumerated:

- Kate Atkinson: "Life After Life"
- Margaret Atwood: "Dancing Girls and other stories" and "Stone Mattress"
- Neil Gaiman: "Neverwhere"
- P. D. James: "The Lighthouse"
- Ruth Rendell: "Collected Short Stories"
- J. K. Rowling: "The Casual Vacancy"
- Val McDermid: "A Place of Execution"

## 3. Proposed CGR-DL Approach

The distinct nature of the presented methodology is described in the following, in opposition to the state of the art.

### 3.1. State of the Art

The traditional approach to the authorship attribution task consists of the pipeline: extraction of stylometric characteristics, feature selection and classification by machine learning techniques. Obviously, feature processing is crucial for a subsequent accurate correspondence between the

stylometric traits and the corresponding author. These features are lexical, character, syntactic, semantic and application specific (literary works, blogs, messages on social media) [1,7]. Regarding the machine learning classifiers, the common choices are support vector machines and neural networks. An intuitive representation of the main steps that are involved in authorship attribution is illustrated in Figure 1.

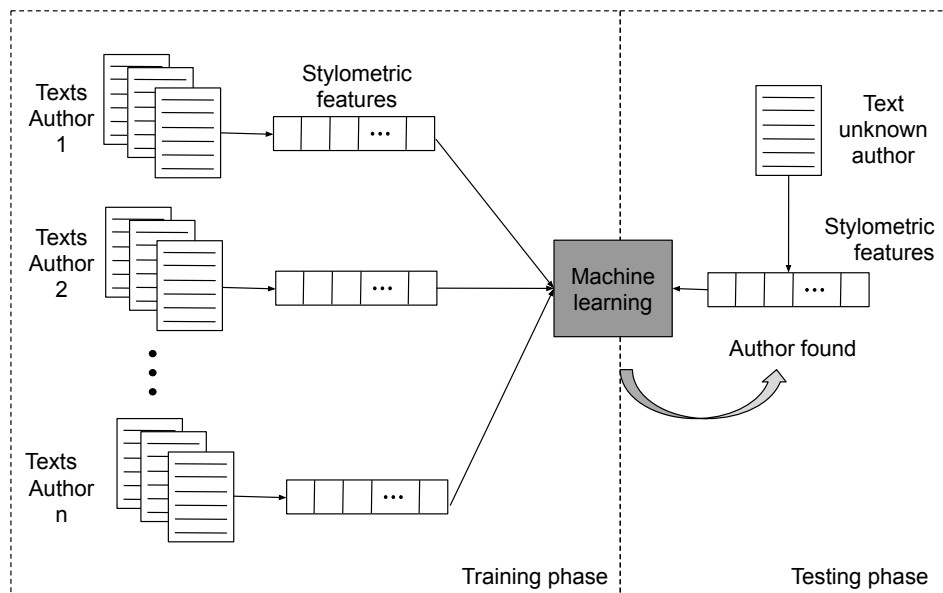

**Figure 1.** General workflow of an authorship attribution procedure.

Recent example entries of the former method target the n-grams directly [8] or following text distortion for masking information unrelated to the style of the author [9], while the latter is applied for a continuous representation of the n-grams [10]. Tensor space models as put forward in Reference [11] require fewer parameters as compared to the traditional vector representation and use a generalized support vector machine that is able to handle tensors. In Reference [12], orthogonal similarity patterns are extracted by combining lexical and stylistic characteristics with modality specific similarity relations. The bag of local histograms proposed in Reference [13] keep track of the term occurrence frequencies across various positions in the document. Again, a support vector machine represents the classifier of choice and the most prolific setting was obtained when using a diffusion kernel. The results obtained by these methods for the CCAT data set will be presented in the experiments section. In the inspiring comparative study [7], instance-based methods that use characters, words, other simple lexical measurements are used for the IMDb task besides a method that employs part-of-speech tags. They also combine different language models and unions of feature types, providing thus a great variety of methods applied for the IMDb task.

DL architectures are also employed for authorship identification. Autoencoders are used in Reference [14] for feature extraction from variable-sized n-grams, followed by SVM classification. Authorship recognition through gated recurrent unit and long short-term memory networks at sentence and article level is done in Reference [15]. The authorship of messages on social media is investigated by convolutional neural networks (CNN) applied on character n-grams in Reference [16]. Such small texts are also treated as uni-dimensional signals to be fed to deep networks in Reference [17]. Generative adversarial network (GAN) approaches are used for the representation of natural language [18], although not for the AA task. In this respect, DL is successfully applied for the interpretation of text from natural language [19,20], as well as from protein sequences [21,22].

The method proposed in this paper is based on a completely different conceptual view of authorship attribution. CGR transfers the text documents to images and these are given to a DL model for classification. A first attempt is made in Reference [4] with the same CGR encoding but

with classification made by traditional machine learning approaches. Recent studies indicate that machine learning classifiers can be successfully used to classify DNA sequences that are produced via CGR [23,24], and these encourage us to explore the CGR suitability for text from natural language.

### 3.2. Chaos Game Representation

The purpose of CGR was to visualize DNA sequences [25,26]. The technique is inspired from chaotic dynamics and it allows the visualization of patterns in structures. In Reference [25], the CGRs obtained from certain group of genes (e.g., human chromosome 11, bacteriophage lambda, Human T-cell Lymphotropic virus) are represented and they reveal remarkable differences. We next present how the method is applied for DNA sequences and later discuss in Section 3.4 how the methodology is adapted for encoding texts.

The alphabet of the DNA sequences contains only four characters that is, A, C, G and T and the representation is initialized with a square that has in its corners these four letters. The square serves as the canvas that will be filled with points for each character of the sequence and it will eventually represent the final CGR image. There is a starting point in the middle of the square. Each nucleotide is represented as the middle point between the previous point and the square corner with its assigned letter. The process continues until the last nucleotide of the sequence is reached. The square image represented with a size of $2^k \times 2^k$ represents every pixel as a different $k$-mer [25]. The total number of $k$-mers from the DNA that are located in the same position of the square reflects the gray level of that pixel.

Any pattern in the DNA sequence triggers a visual pattern in the CGR. The size of the CGR image limits the detail that may be shown. In the implementation used in the current study, a $k$ equal to 7 is considered for the $2^k \times 2^k$ CGR image. However, if $k$ is increased, a finer structure is revealed, but with a cost of a higher resolution image.

### 3.3. The CGR-DL Algorithm

Algorithm 1 shows intuitively how the entire procedure takes place. Regardless of their belonging to the training/validation/test set, the text documents are transformed to images using the CGR (lines 1–6). The text needs some pre-processing to be able to apply CGR; as mentioned in the previous subsection, CGR is applied to sequences that contain an alphabet with only 4 characters. Hence we chose to represent each character in the original text through a pair of base-4 digits, which allows the utilization of maximum 16 distinct characters. The details are described in Section 3.4. Next, the DL approach is applied on the obtained images (lines 7–8) and the details about the architecture are discussed in Section 3.5.

---

**Algorithm 1:** The CGR-DL model.

**Data:** Text documents separated into training-validation-test samples
**Result:** Classification accuracy for the test samples
**for** *each document in the training/validation/test set* **do**
    Reduce the alphabet to 16 characters;
    Encode each character as a pair of base-4 digits;
    Transform the base-4 digits into binary representation;
    Obtain a CGR representation;
**end**
Train DL architecture on the training CGR images using the validation accuracy as criterion to
  select the best model;
Apply DL model on the CGR images from the test set;

---

*3.4. From Text to CGR Images*

The alphabet from regular text documents has to be transformed to be able to apply the CGR representation on it. A base-4 alphabet is aimed for representing the text in a manner that can be used by CGR. Another goal we had in mind is to not extend the base-4 too much as compared to the initial English alphabet representation. Accordingly, some measures were taken to reduce the initial representation of the texts:

- Case sensitivity was not used.
- Some characters are grouped in equivalence classes, as shown in Table 1.
- Each occurrence in text documents of a character from a distinct equivalence class is replaced by the corresponding base-4 pair of digits.

**Table 1.** Each letter from class column is replaced by the pair of digits.

| Class | Base-4 | Class | Base-4 |
|:-----:|:------:|:-----:|:------:|
| {g, h, j} | 00 | {e} | 20 |
| {i, y} | 01 | {n} | 21 |
| {t} | 02 | {\t, \n, (, ), -, +, [, ], 0, 1, ..., 9, ?, !, :, ;, „ .} | 22 |
| {m} | 03 | {u} | 23 |
| {l, r} | 10 | {b, d, p} | 30 |
| {a} | 11 | {f, v, w} | 31 |
| {␣} | 12 | {c, k, q, x, z} | 32 |
| {s} | 13 | {o} | 33 |

The equivalence classes from Table 1 are chosen using a trial and error approach using several criteria for evaluation. One regards the creation of a good balance between the number of occurrences for each of the 16 base-4 combinations in regular documents. A different criterion regarded the balance between the base-4 encodings that start with the same digit, for example, the number of times that 00, 01, 02 and 03 representations appear in a document should not be a lot larger (or smaller) than the number of times that 10, 11, 12 and 13 representations appear in the same document, and idem for the pairs that start with 2 and with 3. Last, but not least, the classification method in Reference [27] was tested on the obtained CGR representations for CCAT10 and these substitutions led to the highest accuracy as opposed to various other encodings tried.

Figure 2 illustrates a step by step transformation of the first six verses from The Raven by Allan Poe: the alphabet from the text in the first rectangle is reduced to the one in the second and then the codifications in Table 1 are applied for each character to reach the values from the third point. Finally, these are transformed to the CGR at the bottom of the figure.

When the documents are small, one CGR can be created from a single text. If the documents are very large, for example, they are books rather than articles, they can be cut into smaller chunks and each such piece can be used to generate a CGR image. In Reference [27] it is shown that chunks starting from 3000 base-4 characters achieve already good results for the classification accuracy. Figure 3 shows 4 images for CGR obtained from chunks of 4000 base-4 characters size of distinct authors. The different representations are hardly distinguishable for the human eye.

A representation of texts via CGR was previously shown in Reference [25], but without further processing on the image. Moreover, the representation took into consideration only four letters, as opposed to the encoding proposed here.

| 1. Input Text - The Raven by Edgar Allan Poe | 2. Result after character substitution |
|---|---|
| Once upon a midnight dreary, while I pondered, weak and weary,<br>Over many a quaint and curious volume of forgotten lore—<br>　　While I nodded, nearly napping, suddenly there came a tapping,<br>As of some one gently rapping, rapping at my chamber door.<br>"'Tis some visitor," I muttered, "tapping at my chamber door—<br>　　Only this and nothing more." | once ubon a mibniggt breari0 wgire i bonbereb0 weac anb weari00<br>ower mani a cuaint anb curious worume ow worgotten rore00<br>　　wgire i nobbeb0 nearri nabbing0 subbenri tgere came a tabbing00<br>as ow some one gentri rabbing0 rabbing at mi cgamber boor00<br>00tis some wisitor00 i muttereb0 0tabbing at mi cgamber boor00<br>　　onri tgis anb notging more00 |

| 3. Text converted to base 4 encoding |
|---|
| 33 21 32 20 12 23 30 33 21 12 11 12 03 01 30 21 01 00 00 02 12 30 10 20 1 1 10 01 22 12 31 00 01 10 20 12 01 12 30 33 21 30 10 10 20 30 22 12 31 20 11 32 12 11 21 30 12 31 20 11 10 01 22 22<br>33 31 20 10 12 03 11 21 01 12 11 12 32 23 11 01 21 02 12 11 21 30 12 32 23 10 01 33 23 13 12 31 33 10 23 03 20 12 33 31 12 31 33 10 00 33 02 02 20 21 12 10 33 10 20 22 22<br>12 12 12 12 31 00 01 10 20 12 01 12 21 33 30 30 20 30 22 12 21 20 11 10 10 01 12 21 11 30 30 01 21 00 22 12 13 23 30 30 20 21 10 01 12 02 00 00 20 12 32 11 03 20 12 11 11 02 11 30 30 01 21 00 22 22<br>11 13 12 33 31 12 13 33 03 20 12 33 21 20 12 00 20 12 01 12 10 11 30 30 01 21 00 22 12 10 11 30 30 01 21 00 12 01 11 02 12 03 01 12 32 00 11 03 30 20 10 12 30 33 33 10 22 22<br>22 22 02 01 13 12 13 33 03 20 12 31 01 13 01 02 33 10 22 22 12 01 12 03 23 02 02 20 10 20 30 22 12 22 01 11 30 30 01 21 00 12 11 02 12 03 01 12 32 00 11 03 30 20 10 12 30 33 33 10 22 22<br>12 12 12 12 12 12 12 12 12 12 12 33 21 10 01 12 02 00 01 13 12 11 21 30 12 21 33 02 00 01 21 00 12 03 33 10 20 22 22 |

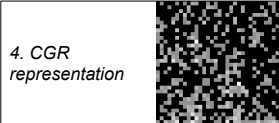

4. CGR representation

**Figure 2.** Example of the transformation of a small text to chaos game representation (CGR).

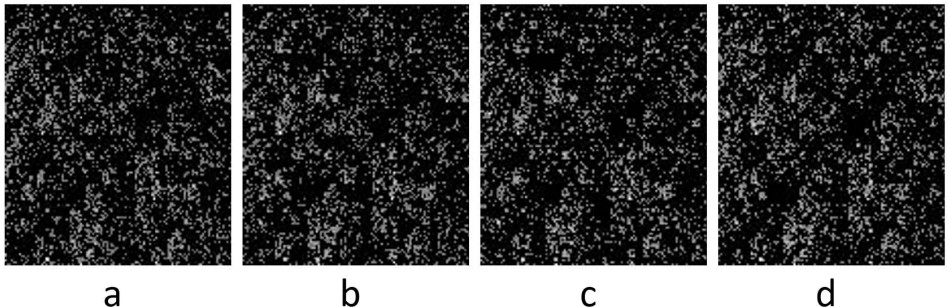

a　　　　　　　b　　　　　　　c　　　　　　　d

**Figure 3.** Examples of CGR images for a small section from Kate Atkinson, "Life After Life" (**a**), Margaret Atwood, "Dancing Girls and other stories" (**b**), Neil Gaiman, "Neverwhere" (**c**) and P. D. James: "The Lighthouse" (**d**).

*3.5. Deep Learning Architecture for CGR Images*

There have been various convolutional neural network (CNN) architectures tried in pre-experimental testing that led us to the conclusion that the CGR images need a very simple DL approach. This is represented in Figure 4.

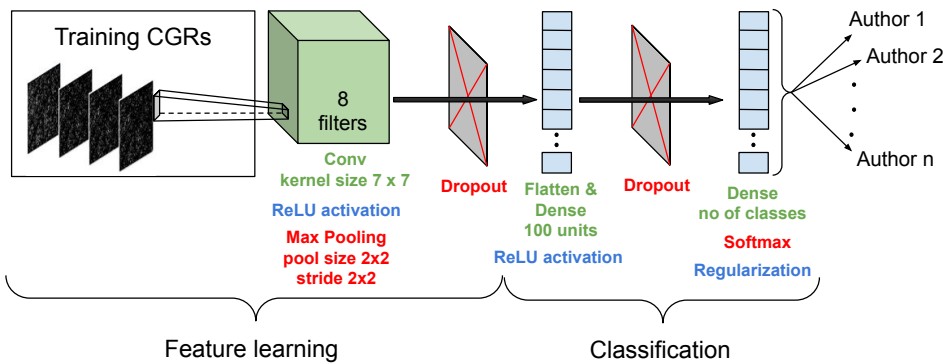

**Figure 4.** Overview of the simple deep learning (DL) architecture used for distinguishing CGR images.

## 4. Experimental Results

There are 3 scenarios tried in the current experimental section. Firstly, some tests are performed on a small data set like CCAT. This allows various settings to be tried since the runs complete in reasonable running time. Secondly, a large data set like IMDb can be used for observing how the proposed CGR-DL tandem performs with such data sets. Lastly, the model is applied for the notorious Rowling case with the 7 candidate authors mentioned in Section 2 used for training.

### 4.1. Case Study: CCAT10

CCAT10 represents a relatively small data set that allows us to study several values for the CNN parameters and even try various architectures. Next, the tasks for the current experiment are defined and its setup and results are described and further discussed.

#### 4.1.1. Tasks

The following tasks will be achieved in the current experiment:

- Parameter tuning is performed for choosing proper values, especially for the dropout rates.
- Several manners to extend the training data set are investigated with the purpose of improving the final classification accuracy on the test set. One of them involves the use of the GAN framework for generating more samples.

#### 4.1.2. Pre-Experimental Planning

Our CNN architecture that was firstly used contained another convolutional layer, in addition to the current one illustrated in Figure 4. It was located just before the *flatten* layer, it had 16 filters and the kernel size of $5 \times 5$, using also a ReLU activation, and was followed by a max pooling layer like the one described in the figure. Although the CGR images clearly do not possess characteristics from pictures from the real-world, transfer learning was also tried from a VGG16 and a ResNet50 with a various number of layers frozen. It was observed however that the more complex architectures led to overfitting and the one described in Section 3.5 proved to be more suitable for this task.

#### 4.1.3. Setup

One of the main problems that are raised by the CCAT10 data set is represented by the reduced number of samples: there are 10 authors (classes), each having 50 samples for training and another 50 for test. A validation set has to be obtained from the training data and thus 20 samples of these are selected for this purpose. Consequently, the CNN will need to be trained only on 300 CGR images in a problem that has 10 classes. Besides the usual split, we also apply a Monte Carlo cross-validation in which the same amount of samples are used for the test set, that is, 50%, while the training data consists of 30% and the validation contains the rest of 20%. The samples are randomly distributed in 5 distinct runs and the average result is reported in this case.

The image sizes are $128 \times 128$ pixels. A batch size of 32 is considered. All possibilities from {0.1, 0.3, 0.5, 0.7, 0.9} are tried for the dropout rates. For preventing overfitting, an L2 kernel regularizer with a value of 0.01 and an L1 activity regularizer with the same value are utilised. A RMSprop optimizer with default parameters is used. When selecting the best model from validation, the accuracy is monitored over the epochs. The usual stop condition contained 1000 epochs, but an early stopping criterion intervenes if there are 20 iterations without any improvement in the validation accuracy.

When the number of images is very small for a CNN to train on, one usual approach is to generate artificial data by applying augmentation to the existing samples. We considered in this respect a rotation range of 20 degrees, width and height shift ranges with a ratio of 0.2, and allowed horizontal flips.

A second scenario was tried to artificially generate more training and validation CGR images from the texts. In this respect, each author had training samples unified in one document, this was

further divided into 500 parts, shuffled randomly, and then a partitioning into groups of 10 led to 50 restitched examples per author. The entire process was repeated 5 times and this resulted in 250 distinct documents for each author, which further correspond to 250 CGR images. Additionally, the original documents are kept, so in total there were 300 CGR images for every author in the training set. From these, 100 were taken out to represent the validation documents.

A third option to artificially increase the number of training samples involves the use of GAN approach. Although the methodology is rather novel, it was successfully applied in various fields of medicine [28,29], for natural images [30], for face portraits [31], as well as for areas that do not involve images, like speech conversion [32]. In brief, a GAN is composed of a generator and a discriminator. The latter learns to differentiate whether the data it receives comes from a real data distribution or it is produced by the generator. Naturally, the generator learns to produce better samples to confuse the discriminator. We implemented a framework similar to the DCGAN (deep convolutional GAN) in Reference [29], where both the discriminator and the generator are CNNs. More details about the settings of the GAN architecture are discussed in Appendix A. The GAN is trained for 200 epochs and for each class there are 30 images that are artificially created this way, hence the training set is doubled.

### 4.1.4. Results & Visualization

In the first two artificially augmented settings, the architecture was not able to learn anything from the artificially generated images, as the classification accuracy was below 20% in all settings tried for the dropout rates. Therefore, these two artificial image augmentation ideas for increasing the number of training samples were abandoned.

The proposed CGR-DL reached an accuracy of 82% on the test set in the best configuration found in validation without any augmentation. The approach that involves the GAN for generating new images led in the best case to a test classification accuracy of 83%. The result stands well against the results from other approaches, as shown in Table 2. More detailed per class results like precision, recall and F1-score are reported in Table 3. The training running time took 76.5 s on a PC with an i7 processor, 16 GB RAM and a GeForce GTX 1650 GPU. The DL code is written in Python, using TensorFlow, and it runs using the GPU. The source code can be accessed at https://github.com/catalinstoean/CGR-DL.

When the Monte Carlo cross-validation is used, the average test accuracy reaches 85.52%.

**Table 2.** Results of the proposed technique and of other methods on the CCAT10 benchmark data set.

| Method | Classification Accuracy |
|---|:---:|
| CGR-DL | 83 |
| FCGR-SVD-NN average [4] | 86.5 |
| FCGR-SVD-NN aggregated [4] | 85.9 |
| FCGR-LR [27] | 82.2 |
| SVM bag of local histogram [13] | 86.4 |
| SVM affix+punct [8] | 78.8 |
| Adam n-gram char (1,2) [10] | 77.8 |
| Adam n-gram char (2,3,4) [10] | 74.8 |
| SVM 3-grams [11] | 80.8 |
| Baseline n-grams [9] | 80.6 |
| MSMF+FLF SVM [12] | 78.8 |

**Table 3.** Precision, recall and F1-score results for each class in turn for the CGR-DL on CCAT10. The last two rows show the macro and weighted averages.

| Label/Measure | Precision | Recall | F1-Score |
|---|---|---|---|
| Alan Crosby | 85.7 | 84.0 | 84.8 |
| Alexander Smith | 84.7 | 100.0 | 91.7 |
| Benjamin Kang Lim | 84.1 | 74.0 | 78.7 |
| David Lawder | 90.0 | 18.0 | 30.0 |
| Jim Gilchrist | 100.0 | 98.0 | 99.0 |
| John Mastrini | 84.8 | 78.0 | 81.2 |
| Marcel Michelson | 97.9 | 94.0 | 95.9 |
| Mure Dickie | 80.0 | 96.0 | 87.3 |
| Robin Sidel | 95.9 | 94.0 | 94.9 |
| Todd Nissen | 54.7 | 94.0 | 69.1 |
| Macro avg | 85.8 | 83 | 81.3 |
| Weighted avg | 85.8 | 83 | 81.3 |

4.1.5. Discussion

There were many measures taken to prevent overfitting. While some could not be successfully fructified, like performing data augmentation through traditional means, others were applied and the obtained results can be considered solid among the state-of-the-art outputs for this data set. Among the successful measures applied, we simplified the model, adopted early stopping of the training, we used regularization and dropout rates and we employed GAN for producing new data.

An in-depth search for the dropout rates is made by varying their values and observing their relationship with respect to the validation accuracy and loss. The values that were chosen for these parameters were of 0.3. The details of the experiment can be read in Appendix B. We also tried then several values for the number of filters (in powers of 2, from 4 to 32) and for the kernel sizes (3, 5, 7 and 9) of the CNN. We found out that the results had a decline for 32 filters and for kernel sizes of $9 \times 9$, while for the others the results remained relatively consistent for all cases. The values illustrated in Figure 4 were kept.

The results from the second and third row from Table 2 are obtained by the same CGR representation as the one used herein, but further dimension reduction via singular value decomposition was achieved and the obtained vectors got classified by a neural network. Additionally, 10 runs were considered for each test sample and the average and aggregated results refer to the probabilities for each class where the test samples were assigned. The average and the aggregated results led to improved classification accuracy. In a previous attempt [27], linear regression was used after the CGR encoding and the results were very close to the current ones using DL. Still, the current result remains remarkable through the fact that it is competitive, despite the fact that the number of training images was very small. The best result in Reference [13] remains controversial since there were two works [10,33] published afterwards that claimed that these outputs could not be replicated. The other classifiers present in the comparison mainly used n-grams of different types in conjunction with SVM or Adam and the results were similar or even inferior to the proposed method.

In an attempt to create a similar training-test split with the one used by the CGR-DL, a training set that contains only the first 30 articles for each distinct author (instead of all 50 items) is considered for the FCGR-SVD-NN average [4], the method that led to the best result for the current data set. Having only 300 samples in the training, instead of 500, influences the singular value decomposition which can thus have only 300 values in the represented vectors. This, combined with the fact that the training set is considerably smaller, further leads the FCGR-SVD-NN method to an accuracy of 82%, which stands in line with the output of the proposed CGR-DL method.

The result obtained from the random splits into training-validation-test is surprisingly good as compared than the one obtained when the usual training and test sets are used (even with augmentation via GANs). We assume that the training and test sets proposed in the data set description represent a

more difficult task, perhaps because they are put in chronological order starting from the first article in the training set and up to the last one in the test set. If this is the case, then by randomizing the entire data set, the model gets an advantage by having in the training data documents written by the authors later.

### 4.2. Case Study: IMDb62

The IMDb62 contains a large number of samples which can be transformed into CGR images and the CNN can learn particularities from many instances.

#### 4.2.1. Task

The aim of this experiment is to use the information found in the first experiment and, if possible, further adapt the values for the parameters to boost performance for the IMDb62 data set.

#### 4.2.2. Setup

The data set contains 62 classes and the split of the samples for training-validation-test sets is 80%–10%–10%. The total amount of samples is 79,550 and the CGR images occupy on disk 2.73 GB. Naturally, the three sets are pairwise disjoint. The same image sizes of $128 \times 128$ pixels and batch size of 32 are considered.

The running time is considerable higher for the current task, so it prohibited us for testing too many values from the input parameters. In this respect, we used some of the findings from the previous experiment and tried a fine tuning at a lower scale. Since the combinations for dropout rates proved to be decisive for the CCAT10 data and usually lower values for both rates, but especially for the second one, led to better results, the possibilities for the first one were taken from {0.1, 0.3, 0.5}, while for the second the options tried were {0.1, 0.3}. The reported result is achieved by the best model chosen from validation at the end of the running time, with early stopping condition of 20 iterations. Each configuration was tested for one training of the CNN. The setting that proved to be the best, that is, for the first dropout rate 0.5, and for the second one 0.3, was repeated 5 times and the final result is calculated as its mean value. Additionally, 5 fold cross-validation is applied with the following setup: the samples in the data set are shuffled and then they are split into 80% training, 10% validation and 10% test. There are 5 distinct splits and the reported classification accuracy is obtained from the average of these 5 runs.

#### 4.2.3. Results & Visualization

Figure 5 depicts the training and validation accuracy during the model learning process in the left plot, while the right one shows the loss values for the two sets, when applying the model at every epoch.

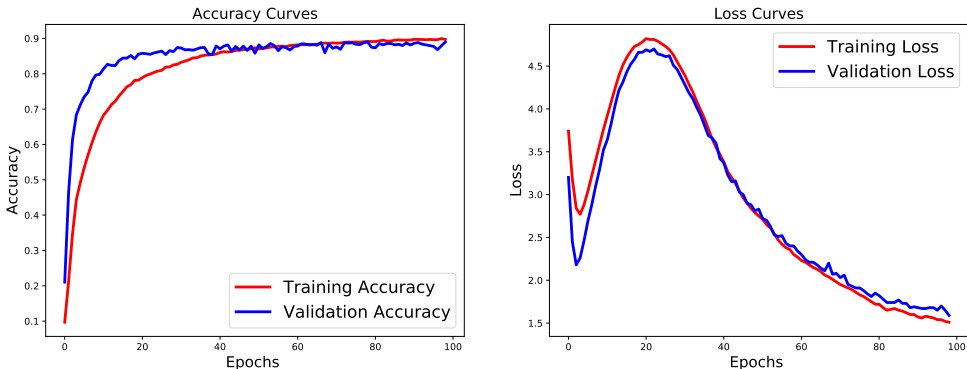

**Figure 5.** Accuracy and loss evolution during the training and validation of the model.

The mean test classification accuracy when the proposed methodology is applied to the same split training-validation-test sets in 5 repeated runs was of 90.57%, while the model reached in average 90.28% on the validation samples. When the 5 fold cross-validation is applied, the mean test accuracy result is of 89.9%. The training running time took in average 93 min and the mean number of epochs was of 81 due to the early stopping criterion.

### 4.2.4. Discussion

The search for proper values for dropout rates is discussed in detail in Appendix B.2. The chosen values for the two are 0.5 and 0.3. The plots in Figure 5 prove that no overfitting takes place for the IMDb62 data set. While the accuracy for both validation and test sets had a very small variation, the running time ranged from 121 min to 280. This large difference happened due to the additional early stopping condition that led to large discrepancies between the total number of epochs of the CNN procedure. The low variation in the classification accuracy proves that the model is robust and probably a lower number than 20 iterations for the early stopping would lead to a similar result, but at a significantly lower computational cost. Moreover, the result remains almost unchanged even when 5 fold cross-validation is applied, underlining the stability of the model.

A good recent study that contains results from several methods applied for the IMDb data set can be found in Reference [7]. There are various characteristics calculated along many classifiers applied on the selected features and the results have a large range, going up to 95.9 in the best case when three different methodologies are combined. There are 5 methods that have the results below the one of the proposed approach, while 7 of them are better. However, 5 of all 12 have the outputs within a range of $\pm 2\%$ from our result. Still, none of the other approaches from Reference [7] used a validation set and the splitting was different from the one herein: while in the current study the splits for training-validation-test are 80%–10%–10%, in Reference [7] it was training-test 70%–30%.

### 4.3. Case Study: Find Rowling Out of 7 Candidates

The data connected to this experiment is introduced in Section 2. The details of the experiment are subsequently described.

### 4.3.1. Task

For this experiment, it is aimed to learn the writing styles of 7 authors starting from their novels having similar subjects and then apply the model on the novel written under the pseudonym Robert Galbraith to uncover the real writer.

### 4.3.2. Setup

As mentioned in Section 2, there are 7 authors of the same types of novels that are considered for training the CGR-DL model. Each novel is separated into fixed text chunk sizes of 3500 characters and these parts are converted to CGR images. The number of chunks for every novel in the training and validation sets are given in the first 3 columns in Table 4.

### 4.3.3. Results & Visualization

Table 4 shows the validation results (precision, recall and F1-score) for each class in turn. The overall validation classification accuracy is 99.4%.

The test accuracy is of 85.9%. There are 9 chunks attributed to Atkinson, 16 to Atwood, 2 classified as pieces written by James, 6 by McDermid and the rest of 201 as Rowling. The manner in which the chunks are classified, in the order they appear in the novel, are illustrated in Figure 6.

**Table 4.** Precision, recall and F1-score results for each class in turn for the validation set. The last two rows show the macro and weighted averages.

| Label/Measure | Chunks Training | Chunks Validation | Precision | Recall | F1-Score |
|---|---|---|---|---|---|
| Kate Atkinson | 112 | 111 | 100 | 100 | 100 |
| Margaret Atwood | 74 | 73 | 100 | 97.3 | 98.6 |
| Neil Gaiman | 79 | 78 | 100 | 98.7 | 99.4 |
| P. D. James | 97 | 96 | 97.9 | 99 | 98.4 |
| Val McDermid | 122 | 122 | 100 | 99.2 | 99.6 |
| Ruth Rendell | 183 | 183 | 100 | 100 | 100 |
| J. K. Rowling | 131 | 130 | 97.7 | 100 | 98.9 |
| Macro avg | - | - | 99.4 | 99.2 | 99.3 |
| Weighted avg | - | - | 99.4 | 99.4 | 99.4 |

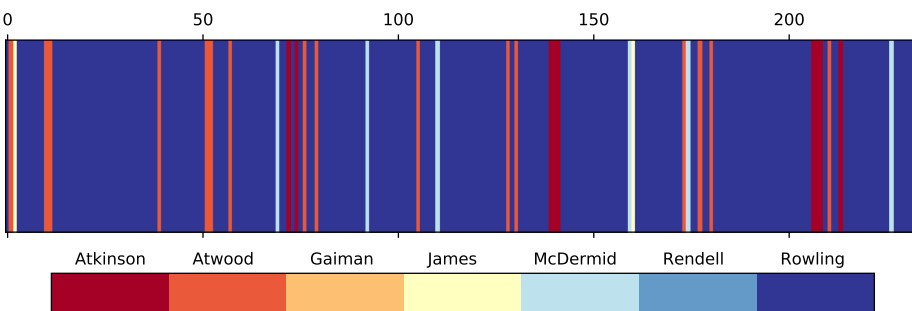

**Figure 6.** Classification of the 234 chunks from *The Cuckoo's Calling* using the trained CGR-DL model.

### 4.3.4. Discussion

The validation accuracy is almost perfect, while the test one is of 85.9%. This might appear like overfitting, but one has to acknowledge that the validation is made on chunks coming from the same novels as the texts in the training set, while the test cases come from a completely different novel. Nevertheless, having 85.9% of the chunks attributed to Rowling and the next choice being Atwood with 6.8%, the decision clearly outlines that Robert Galbraith is the pseudonym used by J. K. Rowling. Although the task is popular from the international press [34,35], there is no benchmark established for this case. Still this is a real-world application of AA and the initial result made the author recognize the fact that she wrote under that pseudonym.

## 5. Conclusions and Future Work

Chaos game representation is used in the current work to transform text documents into $128 \times 128$ pixel images which are further learned by a CNN model. This is next used to determine the class of other images that are transformed using the same CGR procedure.

There are three questions that are suggested in the introduction of the article and for which responses were generated based on the outcomes from the experiments. Based on the three test cases, one can surely state that DL is capable of extracting useful information from the CGR images. Moreover, as observed from the CCAT10 case, the architecture of the CNN should be kept simple to be able to learn the characteristics encoded in the CGRs for different authors. This probably happens because the representation of the CGR images is relatively simple, and there are not so many particularities that need to be taken into account as in the case of real-world images.

As concerns the second question, by comparing the results from the CCAT10 and IMDb62, we conclude that more images lead to more consistent results. This deduction does not refer necessarily to the final classification accuracy and its integration within the results from the other state-of-the-art methods, but rather by comparing the validation outputs with the test results: while for small training and validation data sets the model would not generalize well for unseen test samples, when larger sets are available, the model does not overfit. This can be compared in the RSM plots for accuracy and loss,

where the results are better for the validation set for CCAT10, while they are almost equal for IMDb62 to the test outcomes, hence the model generalized better.

As regards the third research question, the generation of additional CGR images by straightforward means did not lead to good results. One of the tried possibilities regarded the usual augmentation options, like rotation, shifts, flips, but these made the CNN model unable to learn anything from the newly created data. The second option tried refers to randomizing the training texts several times with the aim of creating new CGR images from each such scrambled text data. These newly obtained CGR data were added to the ones generated in the usual way. In this case the model was able to learn features particular to each author, but the test classification accuracy remained inferior to the case when no additional data was added. The third manner of producing new samples involved the use of GAN. This led to doubling the training data and the CGR-DL procedure learned the characteristics of each author better than when only the original data was considered.

The CCAT10 data set is especially used as a testbed for searching parameter settings that would work for the other data sets, as well. However, as fine tuning is also performed at a smaller scale for IMDb62, the findings indicate that the data set has different needs and this suggests that improvements can be achieved. While for CCAT10 a low value for the second dropout rate is recommended, this is kept low for the larger test case, but the promising outputs lay towards its higher values. Further tuning to even higher options might even lead to better results and this remains a task to be explored in the near future, perhaps using metaheuristics like swarm intelligence [36] or monarch butterfly optimization [37], as these were recently successfully applied for such tasks.

It has to be noted that the current results are achieved despite the loss of several characters that are replaced by those representing their equivalence classes. Finding a CGR encoding without loss of information that would eventually be fed to the CNN and improve the classification accuracy is a direction that will be explored in the future. The increase in the size of the CGR images would allow the inclusion of more information, like all the characters in a text and also details like uppercase or lowercase letters. A measure that could lead to further increase in the identification of authors regards the application of a pre-processing step, which could eliminate words carrying information that is irrelevant in describing the style of the author.

**Author Contributions:** Conceptualization, C.S. and D.L.; methodology, C.S. and D.L.; software, C.S. and D.L.; validation, C.S.; investigation, C.S. and D.L.; writing–original draft preparation, C.S.; writing–review and editing, D.L.; visualization, C.S. All authors have read and agreed to the published version of the manuscript.

**Funding:** The research of Catalin Stoean was supported by the grant number 411PED/2020 of the Romanian Ministry of Research and Innovation, CCCDI—UEFISCDI, project code PN-III-P2-2.1-PED-2019-2271, Investigating the influence of optimistic/pessimistic sentiment from text in press releases on stock market trend (INTEREST), within PNCDI III.

**Acknowledgments:** The authors thank the anonymous reviewers whose suggestions helped to improve and clarify this manuscript.

**Conflicts of Interest:** The authors declare no conflict of interest.

## Abbreviations

The following abbreviations are used in this manuscript:

AA　　　Authorship Attribution
CGR　　Chaos Game Representation
DL　　　Deep Learning

## Appendix A. GAN Architecture

Both the generator G and the discriminator D are deep CNNs. The adversarial network assumes an optimization that implies a zero-sum game, where the gain of G corresponds to the loss of D and vice-versa.

For each of the 10 classes of the CCAT data set we trained different GAN models, similarly to Reference [29]. This is necessary since the generator mimics the training images that are provided and the aim of the discriminator is to distinguish between the generated pictures and the real ones. Doing this training for a specific class leads to the creation of a GAN that is able to generate proper images for the specific class that it is trained for.

The architectures used for the generator and discriminator are represented in Figure A1.

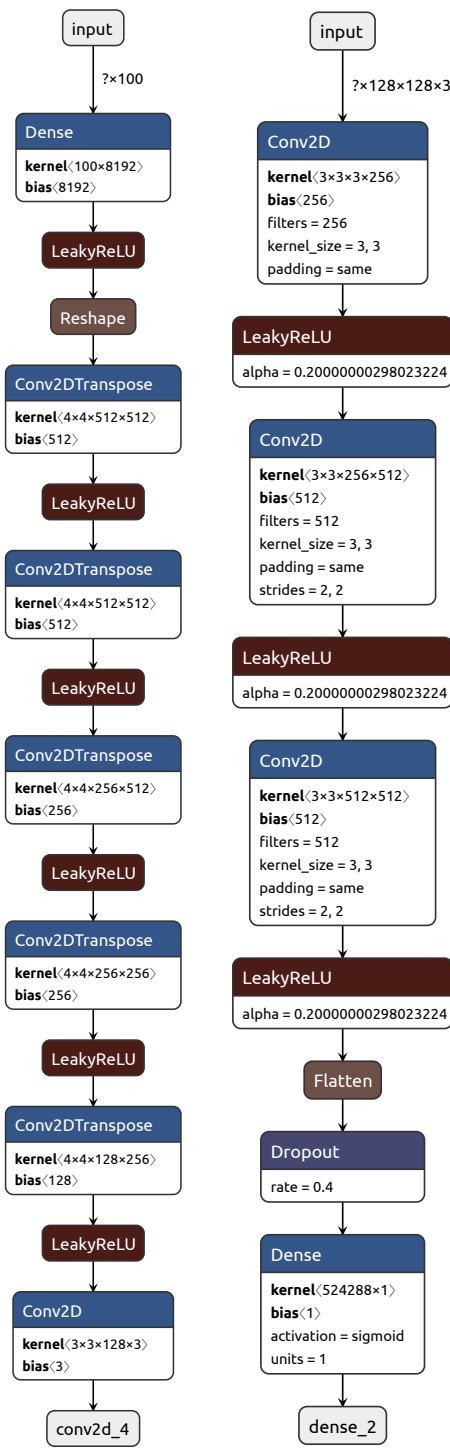

**Figure A1.** The architectures for the generator on the left and discriminator on the right.

The generator takes a point in the latent space, for example, a vector of 100 random numbers and outputs an image of $128 \times 128$ pixels, that is, identical to the CGR initial training images. LeakyReLU is used instead of ReLU with an alpha leak of 0.2, as it is often taken in practice [29]. A stochastic gradient descent with the Adam optimizer is used with a learning rate of $2.0 \times 10^{-4}$ and a momentum of 0.5. The generator starts with a dense layer. The activations are then reshaped to $4 \times 4 \times 512$ feature maps that are further up-sampled by the Conv2DTranspose layer with 512 filters, $4 \times 4$ kernel size and a stride of $2 \times 2$. The same pair LeakyReLU - Conv2DTranspose is applied several times until a convolutional layer of the desired size is reached.

The discriminator has a rather typical CNN architecture that receives an image of size $128 \times 128 \times 3$ and outputs whether the image is real or is produced by the generator. It contains three pairs of convolutional—LeakyReLU layers which are then flattened, a dropout layer is applied and finally a sigmoid activation function establishes the prediction.

## Appendix B. Parameter Tuning for Dropout Rates

Response surface models (RSM) are used for illustrating the relationship between two dropout parameters. These use a set of inputs for the two dropout rates and the calculated results obtained in these points, and further approximate results in their neighborhood. Although RSM are based on approximations and are not totally accurate, they are useful for having an overview of the effects of the combination between the variables.

### Appendix B.1. Dropout Rates for CCAT

Figure A2 illustrates RSM for the validation and classification accuracy for the CCAT data set on the first row. The second row contains the corresponding loss results. While for the first row of plots higher values correspond to better results, it is the opposite for the plots in the second row.

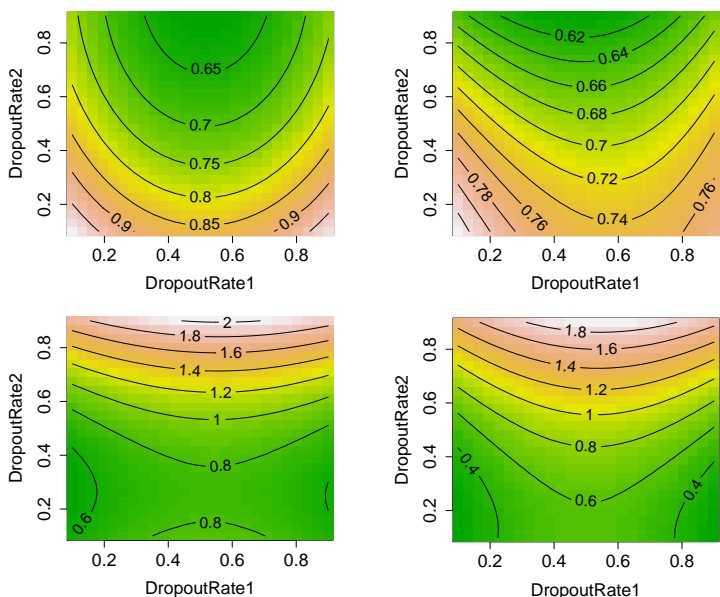

**Figure A2.** RSM are illustrated on the first row for validation and test classification accuracy, when tuning values for the dropout rates. The bottom plots show the corresponding RSM for loss values.

The plots from the first column, corresponding to the validation accuracy and loss, helped us choose an appropriate combination for the dropout rates that led to better results. As it can be observed, the trend is very similar for the test results, that is, second column. The results show that a low dropout rate for the second parameter should be selected and either a low or a larger value for the first dropout

rate should be set, although its value appeared to count less. We chose rates of 0.3 for both parameters for the results reported in Tables 2 and 3.

*Appendix B.2. Dropout Rates for IMDb*

Figure A3 illustrates RSM plots for the validation (left) and test (right) results for the IMDb62 data set. The results from the first row outline the ratios for the classification accuracy, while the second shows the loss.

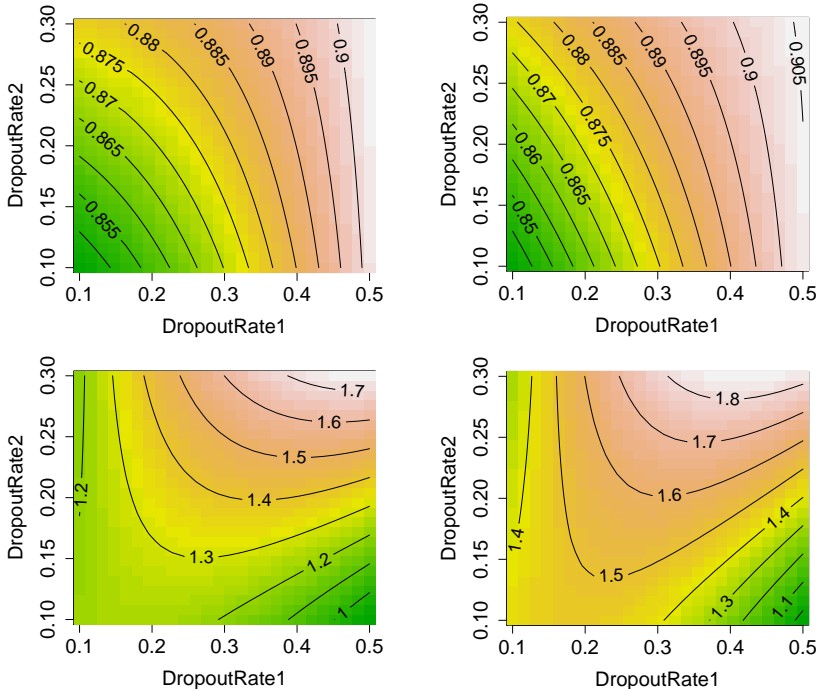

**Figure A3.** RSM for observing the effects of dropout rates on the validation and classification accuracy on the first row and for the corresponding loss values on the second row.

The trends in Figure A3 are very consistent from the validation to the test set with respect to both accuracy and loss. Moreover, almost the same results are achieved for the two sets, indicating that no overfitting takes place for the IMDb62 data set. The RSM plots suggest that the current data set has different necessities as compared to CCAT10, since the hot spots moved here towards the 0.5 and 0.3 values for the two dropouts.

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
