# Peer review of "Author Identification Using Chaos Game Representation and Deep Learning"

_mathematics, doi:10.3390/math8111933_

Round 1

Reviewer 1 Report

Review of “Author Identification using Chaos Game Representation and Deep Learning” Manuscript ID: mathematics-948606 This paper proposed an approach that starts from the character level and uses chaos game representation to illustrate documents like images which are subsequently classified by a deep learning algorithm. The topic “The authorship attribution (AA)” is very interesting to the readers. The authors use a chaos game representation to transform text documents into images and apply a deep neural network to solve the problem. Experiments conducted on three datasets including CCAT10, IMDb62 shows good performance. There’s a great improvement compared with the last version. My questions and concerns have been addressed in the modification. I think this paper is well-written and ready to publish.

Author Response

We are afraid that the review addresses a different article, not our work.

Reviewer 2 Report

This paper presents an interesting work on author identification with a series of effective methods, including CGR, deep learning model, and GAN.
The proposed algorithm is tested on three datasets for different purposes.
In general, the writing is clear, the problem is moderately motivated, and the related work and the proposed method are suitably described.
In Section 3.2, it would be more easily understandable to show a visual example of how CGR creates an image given a sequence as the input.
Table 2 shows that the proposed CGR-DL is inferior to [4] and [13]. Therefore, it would be necessary to compare them in at least one of the other two datasets.

Minor issues:
Line 11: that person has written -> that the person has written
Line 35: and use it to ... -> and using it to
Lines 83-96, Line 97-105: please use the same tense. The former paragraph is written in the present tense while the latter in the past tense.
Line 183: its setup and results ... -> and its setup and results ...
Line 240: with a i7 processor -> with an i7 processor

Author Response

We attach the point-by-point response to the reviewer’s comments.

Reviewer 3 Report

Stoean and Lichtblau present a new text classification pipeline for processing documents in order to assign them to an author. They use a supervised machine learning approach based on the well-known training and test paradigm. In order to process the documents and to extract the features they use a chaos game representation, which enables to encode the texts as images. For performing the classification of the images, they use state-of-the-art deep learning algorithms.
The work is interesting and the methodology seems to be sound. The experimentation shows the validity of the approach and the results are promising.
Additionally, the manuscript has a good structure and is well written, although some minor English corrections are necessary.
Nevertheless, the manuscript needs major improvements before publication, i.e.:

When experimenting the approach, the authors should explain better the adopted sampling procedure. How are training, test, and validation data chosen? What are the proportions?
It seems as the authors have randomly split the dataset with just a single run. If so, the results can be confusing, because they are based on a single draw and it can be a dumb luck.

I suggest also to expand the discussion about future directions at the end of the manuscript.

Author Response

(The authors gave the same response as above.)

Round 2

Reviewer 1 Report

This paper proposed an approach that starts from the character level and uses chaos game representation to illustrate documents like images which are subsequently classified by a deep learning algorithm. The topic “The authorship attribution (AA)” is very interesting to the readers. The authors use
a chaos game representation to transform text documents into images and apply a deep neural network to solve the problem. Experiments conducted on three datasets including CCAT10, IMDb62 shows good performance.

There’s a great improvement compared with the last version. My questions and concerns have been addressed in the modification. I think this paper is well-written and ready to publish.

Author Response

We thank the reviewer for carefully reading the manuscript and for the encouraging comments.

Reviewer 3 Report

I would like to see a 5 fold cross validation test in order to assess the validity of the results.

Author Response

We applied 5 fold cross-validation. We attach a detailed response and the document with the newly added text in blue.

Round 3

Reviewer 3 Report

The authors addressed all my comments and performed the new experimentation with good results

This manuscript is a resubmission of an earlier submission. The following is a list of the peer review reports and author responses from that submission.

Round 1

Reviewer 1 Report

The authors have presented author identification using chaos game representation and deep learning. 

There are several issues that need to be addressed in the paper:

  1. A comprehensive review of the existing approaches need to be included to justify the development of a new approach. 

  2. Need to strongly justify the efficiency and effectiveness of the proposed approach. 

Reviewer 2 Report

The authors proposed a framework for author identification using chaos game representation and deep learning. There are some concerns that cannot meet the quality requirements of Mathematics as follows:

- The overall structure of the manuscript needs to be improved. For example, the discussions for all case studies should be combined together to have a general discussion on the algorithm. Or some literature reviews should be presented in the introduction part...

- Literature reviews are weak. The authors should discuss more related works on this problem, even there are some works that have been done on their selected datasets (CCAT10, IMDb, ...).

- The authors mentioned that they proposed an algorithm namely CGR-DL, but I do not consider it as algorithm. Like the pseudo code in line 102, I cannot understand well the workflow.

- How about the performance results if the authors did not use CNN? The authors have to compare the performance among different traditional machine learning and deep learning techniques.

- What are different among different sub-figures in Fig. 3? It is not presented clearly in the whole manuscript.

- Why could the authors propose that their algorithm worked well? Compared to the other results in Table 2, the classification accuracy was much lower than the previously published works?

- Deep learning (or CNN) has been used in a lot of previous works such as PMID: 31277574 and PMID: 31921391. Therefore, the authors should provide more references to this description.

- The authors should release source codes for reproducing the results.

Reviewer 3 Report

The main criteria for review can include the followings:

The main idea of the paper:

1. Originality and significance of the research
Very interesting work, well researched and well written. But regarding the premise, does an author's style change during his/her lifetime? Did Margaret Atwood's style change throughout her books? Interesting concept to explore. And I'm sure you have thought about multiple authors for one text. Have you thought about incorporating learning text that is plagiarized (multiple authors, and cannot possibly be this current author's style?)

Main idea of the paper is: Learning features of a person's writing style and being able to classify it among other text.

2. Technical and theoretical correctness
Correct

3. Readability of the paper
Readable.

4. Evaluation result
Good use of visualizations, perhaps figure 6 can be made with less contrast? I'm biased, perhaps some readers might like a more vision-impaired appropriate figure, just aesthetics.

5. Scope of the work.
Good scope.

Reviewer 4 Report

This paper proposed a Deep Learning (DL) approach for the authorship attribution (AA) task. It uses Chaos Game Representation (CGR) to encode text into images, and then feed into DNN for classification. The topic is interesting and the method is explained clearly. I have 3 major concerns:
1. The encoding part. Need to try and compare different encoding methods other than CGR, and prove the advantage of CGR over other encoding methods. When converting text to images, we implicitly introduce correlation in pixel intensity, and correlation of pixel location. Natural images have such properties, e.g. pixel intensity 20 is similar to pixel intensity 21, and neighbor pixels are similar. For this reason, slightly changing the pixel intensities or moving pixels for small distance will not affect the classifier output. These kinds of correlation is especially important when we use convolution neural networks. When encoding text to images, we mush prove these correlations in converted images are meaningful and represent some kind of correlations in text. 

2. Data augmentation. The authors only tried basic image augmentation methods: rotation, shifts, flips. However, for images generated from text, they do not have the correlation properties I mentioned above. So these augmentation methods will not work, because the distribution changed. (Think about reversing a poem from end to the beginning, it's no longer a valid poem!) I strongly suggest the authors to try advance data augmentation methods using GANs, i.e. to generate synthesis data from small real dataset. In this way, the generated data will have the same distribution as the real data, and therefore help classification. Two references are suggested for this issue:

[1]. “Nonparallel Emotional Speech Conversion,” INTERSPEECH 2019,  Graz, Austria, September 2019

[2]. "Data augmentation using generative adversarial networks (CycleGAN) to improve generalizability in CT segmentation tasks, Sandfort, K Yan, PJ Pickhardt, RM Summers - Scientific reports, 2019

3. It's better to provide a detailed introduction of Chaos Game Representation in related work. The experiment need to reduce the part of tuning "dropout rate", as this is a minor issue (good for engineering report but not suitable in scientific articles) for deep learning.